# Identification of MicroRNA Expression Profiles Related to the Aggressiveness of Salivary Gland Adenoid Cystic Carcinomas

**DOI:** 10.3390/genes14061220

**Published:** 2023-06-02

**Authors:** Maicon Fernando Zanon, Cristovam Scapulatempo-Neto, Ricardo Ribeiro Gama, Márcia Maria Chiquitelli Marques, Rui Manuel Reis, Adriane Feijó Evangelista

**Affiliations:** 1Molecular Oncology Research Center, Barretos Cancer Hospital, Barretos 14784-400, Brazil; 2Department of Head and Neck Surgery, Barretos Cancer Hospital, Barretos 14784-400, Brazil; 3Life and Health Sciences Research Institute (ICVS), School of Medicine, University of Minho, 4710-057 Braga, Portugal; 4ICVS/3B’s-PT Government Associate Laboratory, 4710-057 Braga, Portugal; 5Sergio Arouca National School of Public Health, Oswaldo Cruz Foundation, Manguinhos, Rio de Janeiro 21040-361, Brazil

**Keywords:** salivary gland adenoid cystic carcinomas, microRNA expression, Brazilian cohort, cancer aggressiveness, FFPE

## Abstract

Adenoid cystic carcinoma (ACC) has been reported as the second most common carcinoma of the salivary glands. Few studies have associated miRNA expression with ACC aggressiveness. In this study, we evaluated the miRNA profile of formalin-fixed, paraffin-embedded (FFPE) samples of salivary gland ACC patients using the NanoString platform. We studied the miRNA expression levels associated with the solid growth pattern, the more aggressive histologic feature of ACCs, compared with the tubular and cribriform growth patterns. Moreover, the perineural invasion status, a common clinicopathological feature of the disease that is frequently associated with the clinical progression of ACC, was investigated. The miRNAs showing significant differences between the study groups were selected for target prediction and functional enrichment, which included associations with the disease according to dedicated databases. We observed decreased expression of miR-181d, miR-23b, miR-455, miR-154-5p, and miR-409 in the solid growth pattern compared with tubular and cribriform growth patterns. In contrast, miR-29c, miR-140, miR-195, miR-24, miR-143, and miR-21 were overexpressed in patients with perineural invasion. Several target genes of the miRNAs identified have been associated with molecular processes involved in cell proliferation, apoptosis, and tumor progression. Together, these findings allowed the characterization of miRNAs potentially associated with aggressiveness in salivary gland adenoid cystic carcinoma. Our results highlight important new miRNA expression profiles involved in ACC carcinogenesis that could be associated with the aggressive behavior of this tumor type.

## 1. Introduction

Adenoid cystic carcinoma (ACC) is the second most common carcinoma of the salivary glands. Approximately 10–15% of salivary gland tumors are ACCs [1,2]. In small salivary glands, ACC is the most common salivary gland malignancy, accounting for ∼50–60% of all cases [3]. ACC is commonly diagnosed in the fifth and sixth decades of life (median age at diagnosis of 54.6 years), with a slight female predominance (60%) [4]. Furthermore, it is usually characterized by a relentless clinical course and a poor long-term prognosis. However, some features are associated with an unfavorable prognosis, such as a solid pattern of growth, a high rate of perineural invasion, local recurrence, and late distant metastasis [5]. ACC has three major growth patterns, i.e., tubular, cribriform, and solid [6]. The solid growth pattern is defined by solid growth in at least 30% of the tumor surface and includes nests and sheets of neoplastic cells [6].

Mutational landscape studies have shown that these tumors display a low mutation burden (approximately 0.3 mutations/Mb) and a low prevalence of mutations in common driver genes, except for Notch pathway genes, which are mutated in nearly 10% of tumors with a solid pattern [7,8,9], and *NOTCH1*-mutant tumors may activate the Notch1 pathway [10]. Therefore, ACC pathogenesis has been associated with a spectrum of complex structural rearrangements, of which a fusion between the *MYB* and *NFIB* genes is the most prominent and a genomic hallmark of the disease [11]. The *MYB-NFIB* fusion occurs mainly due to translocation in t(6;9)(q23;p23), and the truncated transcript is composed of the DNA-binding and transactivation domains of *MYB* and the C-terminal domain of *NFIB* [11]. Furthermore, the *MYBL1-NFIB* fusion has also been detected in ACC, and it is mutually exclusive to the *MYB-NFIB* translocation [12]. *MYB* and *MYBL1* are classes of genes from the same family, i.e., the MYB family of transcription factors, and the same domains are retained in both cases when the fusions occur [12]. Frerich et al. [13] identified three subgroups of ACC tumors based on the expression of *MYB* and *MYBL1* oncogenes, showing a subgroup with significantly worse overall survival. The modulation of *MYB* expression after fusion in ACCs has been associated with mechanisms of enhancer-driven overexpression [14]. However, other mechanisms could be involved, including the *MYB* deregulation attributed to the loss of negative regulatory elements [14]. The lost 3′ untranslated regions (UTRs) of *MYB* contain conserved microRNA binding sites, including miR-15a, miR-16, and miR-150, which play a potential regulatory role, but the underlying mechanism has not yet been fully elucidated [11].

MicroRNAs (miRNAs) are a class of small noncoding RNAs involved in intracellular regulation at the post-transcriptional level of approximately one-third of human genes [15]. These regulatory transcripts are short, single-stranded RNA sequences (approximately 19–23 nucleotides) that modulate gene expression by binding to mRNAs typically within the 3′ UTR, leading to the downregulation of protein expression in a variety of physiological processes and pathological conditions [16]. Moreover, miRNAs are implicated in several carcinogenesis mechanisms and can be of prognostic value, including in salivary gland ACC [17]. Previous studies on miRNA expression in salivary gland tumors have mainly focused on (i) the fusion status [18]; (ii) comparison between ACCs from different sites, such as the breast, salivary, and/or lacrimal glands [19,20,21]; and (iii) miRNA expression in several salivary gland neoplasms, such as pleomorphic adenomas, among others [22,23,24,25,26]. Few studies have focused on tumor progression features, and most of them used cell lines to identify putative metastasis biomarkers [26,27,28]. Recently, Denaro et al. [29] analyzed miRNA expression, comparing benign and malignant salivary gland tumors of several types, and only four cases were ACCs. Therefore, little is known about the role of these miRNAs in the most aggressive salivary gland ACC tumors. Another study by Brayer et al. [30] discusses the comparison of transcription profiles in adenoid cystic carcinoma (ACC) tumor samples from various tissues. The results showed that microRNA expression patterns did not distinguish different types of ACC tumors as effectively as gene expression patterns, suggesting the influence of activated oncogenes on protein-coding genes.

In this study, we performed miRNA expression analysis in salivary gland ACC tumor samples using nCounter Technology (NanoString Technologies, Seattle, WA, USA) to identify differentially expressed miRNAs in distinct ACC patient groups in this tissue. We found 30 differentially expressed miRNAs in the solid growth pattern compared to the tubular and cribriform growth patterns. We also identified 49 miRNAs associated with perineural invasion status. The identified miRNAs are candidates that may distinguish ACC patients with a higher potential for carcinoma aggressiveness.

## 2. Material and Methods

### 2.1. Study Design and Patients

A retrospective sample collection of 19 salivary adenoid cystic carcinoma patients between 1998 and 2012 at Barretos Cancer Hospital was carried out. Table 1 summarizes the main clinicopathological features of the selected patients. All patients received surgery and radiotherapy as treatment, and only one patient (5.2%) received chemotherapy. The age range of the patients was 30–89 years, and the histological growth patterns were 36.8% for cribriform (total of 7), 31.5% for tubular (total of 6), and 26.3% for solid (total of 5). Perineural invasion was present in 11 patients (57.8%) and absent in 8 patients (42.1%). The follow-up time was 10 years, and the disease status at last follow-up showed that 21.0% of patients were alive with the disease, 63.2% were alive without the disease, and 15.8% had died due to the disease (Table 1 and Appendix A). This study was approved by the Barretos Cancer Hospital Ethical Review Committee, which waived the requirement for patients’ written informed consent due to the retrospective nature of the study (nr# 135.352/2012).

### 2.2. RNA Isolation from FFPE Samples

Formalin-fixed, paraffin-embedded (FFPE) ACC samples were obtained from the Department of Pathology of the Barretos Cancer Hospital and RNA was extracted with support of Biobank of the same institution [31]. Tumor cell content (>60%) was assessed based on H&E-stained slides, and RNA was isolated from 10 μm thick tumor sections from biopsy or surgical specimen samples using the RecoverALL Total Nucleic Acid Isolation Kit (Life Technologies) according to the manufacturer’s instructions, as previously reported [32]. In brief, the process is divided into four steps: (i) an initial preparation that includes slide scraping, deparaffinization with xylene, and 100% ethanol dehydration; (ii) protease digestion; (iii) nucleic acid isolation using the filter cartridge, followed by washing; and (iv) DNase digestion with additional washes and elution. RNA quantification was assessed using a Nanodrop 1000 spectrophotometer (NanoDrop Products, Thermo Scientific, Wilmington, DE, USA).

### 2.3. NanoString nCounter miRNA and mRNA Assay

miRNA expression profiling was carried out using the nCounter Human v3 miRNA Expression Assay Kit (NanoString Technologies, Seattle, WA, USA) according to the manufacturer’s protocol, as previously described [33]. An amount of 100 ng of total RNA for all samples was subjected to sample preparation involving multiplexed annealing of specific tags onto the 3’ end of each mature miRNA, followed by a ligation reaction and enzymatic purification to remove nonligated tags. Next, miRNAs were hybridized with probe pairs that comprised biotin-labeled capture probes and fluorescent color-barcoded reporter probes for 21 h at 65 °C. Unhybridized probes were then loaded onto the automated nCounter Sample Prep Station (NanoString Technologies), which performed the magnetic bead-based purification and immobilization of target-probe complexes on cartridges with streptavidin-covered surfaces. Finally, the cartridges were transferred onto the nCounter Digital Analyzer (NanoString Technologies, Seattle, WA, USA) for data collection consisting of digital imaging and direct quantification of the individual fluorescent barcodes. Gene expression analysis was conducted to create miRNA–mRNA networks with targets related to cancer. The nCounter platform (Nanostring technology, Seattle, WA, USA) was used, similar to miRNA expression profiling, and the nCounter PanCancer Pathway panel was used. This panel specifically examines the expression levels of 770 genes, including 606 genes associated with various pathways, 124 driver genes, and 40 housekeeping genes. These genes are involved in 13 molecular pathways, such as Notch, Wnt, Hedgehog, TGFB, MAPK, STAT, P13K, RAS, chromatin modification, transcriptional regulation, DNA damage control, cell cycle, and apoptosis. A custom nCounter Gene Expression Assay (NanoString Technologies) was also performed using the nCounter Vantage 3D RNA:Protein Solid Tumor Assay for Lysates protocol to investigate the expression of specific genes involved in ACC. The protocol followed the manufacturer’s recommendations. Probes to target differences in the expression of genes involved in known fusion processes in ACC were included and designed by Nanostring, such as the *MYB*, *MYBL1*, and *NFIB* genes, which contain important activating mutations related to aggressiveness.

### 2.4. NanoString Data Analysis

The pipeline was similar to that used to analyze the miRNA profiling and PanCancer Panel mRNA targets. NanoString raw RCC files were input into R version 4.0.2 (R Foundation, Vienna, Austria) [34] and analyzed using the NanoStringNorm R package (version 1.1.21) [35]. Then, pre-processing and normalization steps were applied, including filtering due to a lack of probe intensity in at least 80% of cases, probe-level background correction by code-count normalization using the geometric mean parameter, and sample content normalization using the top 10 low coefficient value (CV) probe values. The normalized data were log2-based transformed, and the subsequent differential expression was identified in R using the Linear Models for Microarray Data (limma) package [36]. We used the moderated t-statistics for two-class comparisons between the study groups (solid *versus* other growth patterns and presence *versus* absence of perineural invasion (PNI)). PNI was evaluated from surgical specimen samples on H&E-stained slides by two pathologists independently. An FDR-corrected *p*-value ≤ 0.05 and fold change (FC) ≥ 2.0 were considered. Hierarchical clustering was performed considering Euclidean distance, represented graphically as dendrograms, where the red color indicates upregulation and blue indicates downregulation. The data analysis of the custom nCounter Gene Expression Assay included normalization with known housekeeping genes in ACC (*BID*, *HPRT1*, and *LAT*). The average plus or minus two standard deviations of expression values were used to define positivity of *NOTCH1* gene expression and fusion genes, i.e., *MYB*, *NFIB*, and *MYBL1*.

### 2.5. miRNA Functional Enrichment Analysis

We used DIANA-miRPath v3.0 [37] to further characterize the functional role of the miRNAs identified in the study groups. This tool performs enrichment analysis of multiple miRNA target genes provided by the DIANA-microT-CDS algorithm and/or experimentally validated miRNA interactions derived from DIANA-TarBase v6.0. We selected the experimentally validated miRNA interactions as an analysis parameter and the resulting information from the Kyoto Encyclopedia of Genes and Genomes (KEGG) pathways and Gene Ontology (GO), categorized as GO Slim terms, i.e., cut-down versions of the GO’s ontologies containing its subsets. The statistical significance of an FDR-corrected *p*-value of ≤0.01 was considered after the association with the identified categories according to miRPath analysis. Furthermore, we selected the top categories associated with cancer by features such as cancer progression, proliferation, and apoptosis, among other hallmarks of the disease. The mRNA targets predicted by DIANA-TarBase that were also differentially expressed in the PanCancer panel were selected for the construction of the gene networks. The cytoscape tool (http://www.cytoscape.org, accessed on 13 April 2023) was used to design the miRNA–mRNA networks.. The miRNAs previously associated with ACC were accessed from the curated Human MicroRNA Disease Database (HMDD) v3.2 [38]. In this study, we selected the terms “Carcinoma, Adenoid Cystic”, “Salivary Gland Neoplasms”, and “Lacrimal Adenoid Cystic Carcinoma”.

## 3. Results

We analyzed the expression of 798 miRNAs from 19 salivary gland ACC patients using NanoString technology. We found that there was a female sex prevalence (68.4%), parotid glands were the most frequent carcinoma site (31.5%), and most patients were in clinical stage IV (47.3%) and alive (84.2%) (Table 1 and Appendix A). Moreover, the histological subtypes and perineural invasion were similarly categorized between the two groups (Table 1). Finally, all patients underwent surgery and radiotherapy (Table 1). A non-supervised expression of all the miRNAs analyzed is shown in Appendix A. Overall, the patient samples can be clustered according to the histological pattern. However, the carcinoma sites also show clear separation. A cluster with six patients (five with solid and one with cribriform histological patterns) is related to minor salivary gland carcinoma sites.

### 3.1. Differentially Expressed miRNAs in Solid ACC Compared to Other Growth Patterns

To identify miRNAs that were differentially expressed in the solid pattern compared to the other growth patterns (i.e., tubular and cribriform), we performed the statistical analysis by using a moderated *t*-test, considering fold change ≥ 2.0. We identified a total of 30 significant miRNAs (11 upregulated and 19 downregulated) in the solid growth pattern (Figure 1 and Appendix A). Several mRNA targets related to cancer pathways were identified (Figure 2).

These miRNAs included several important cancer pathways and processes, as obtained by DIANA-miRPath. Forty-four enriched KEGG pathways and forty-two GO processes were compiled as GO Slim terms. Among them, we selected the top three KEGG and GO categories related to cancer. Of the GoSlim results, we identified cell death (GO: 0008219) (27 miRNAs, *p* = 3.12 × 10^−17^), enzyme regulator activity (GO: 0030234) (26 miRNAs, *p* = 5.35 × 10^−14^), and cell–cell signaling (GO: 0007267) (25 miRNAs, *p* = 3.59 × 10^−10^). The KEGG processes obtained were mainly the Hippo signaling pathway (hsa04390) (21 miRNAs, *p* = 6.24 × 10^−6^), the TGF-beta signaling pathway (hsa04350) (20 miRNAs, *p* = 0.0005), and transcriptional misregulation in cancer (hsa05202) (22 miRNAs, *p* = 0.004).

Moreover, the disease association database used (HMDD) showed several associations with cancer. In this analysis, we selected the “Carcinoma, Adenoid Cystic” and “Salivary Gland Neoplasms” categories, which provided five miRNAs of interest: hsa-miR-181d, hsa-miR-23b, hsa-miR-455, hsa-miR-154-5p, and hsa-miR-409 (Table 2).

### 3.2. Differentially Expressed miRNAs in Salivary Gland ACC According to Perineural Invasion

To identify differentially expressed miRNAs in patients with perineural invasion, the moderated *t*-test with fold change ≥ 2.0 was used. A total of forty-nine significant miRNAs were obtained (forty-seven upregulated and two downregulated) according to perineural invasion status (Figure 3 and Appendix A). Several mRNA targets related to cancer pathways were identified (Figure 4).

The miRNAs identified were functionally enriched using the DIANA-miRPath tool. Seventy-five enriched KEGG pathways and fifty-three GO processes were compiled as GO Slim terms. Among them, we selected the top three KEGG and GO categories related to cancer. We found that the most relevant KEGG results were the Hippo signaling pathway (hsa04390) (40 miRNAs, *p* = 9.26 × 10^−8^), the TGF-beta signaling pathway (hsa04350) (38 miRNAs *p* = 9.30 × 10^−7^), and ECM–receptor interaction (hsa04512) (38 miRNAs, *p* = 2.17 × 10^−5^). Of the GO Slim processes, we highlighted cell death (GO:0008219) (43 miRNAs, *p* = 3.12 × 10^−17^), enzyme regulator activity (GO:0030234) (43 miRNAs, *p* = 2.24 × 10^−16^), and cell–cell signaling (GO:0007267) (43 miRNAs, *p* = 1.05 × 10^−15^), all of them in common with the analysis of differentially expressed miRNAs in the solid ACC growth pattern.

Finally, the data provided by the HMDD disease association database with salivary gland neoplasms regarding the perineural invasion analysis are shown in Table 2. According to previous ACC studies, we found six miRNAs related to the disease (hsa-miR-29c, hsa-miR-140, hsa-miR-195, hsa-miR-24, hsa-miR-143, and hsa-miR-21) (Table 2).

### 3.3. MicroRNA Expression between Tumors Arising in Major and Minor Salivary Glands

The miRNAs identified in salivary gland ACC according to growth patterns and perineural invasion were compared with those differentially expressed between major and minor salivary glands (Figure 5). The results show the miRNAs are potentially related to each site of origin and aggressiveness. Overall, one miRNA was upregulated in the PNI and major salivary glands (Figure 5A). In the solid growth pattern, seven upregulated miRNAs were found in minor salivary glands (Figure 5C), and eleven downregulated miRNAs were found in major salivary glands (Figure 5D).

## 4. Discussion

ACCs constitute 10–15% of salivary gland carcinomas, corresponding to the second most common carcinoma of the site [1,2]. ACCs most often occur in the minor salivary glands and the submandibular gland, and less commonly in the sublingual and parotid glands [41,42]. However, the parotid gland was the most common carcinoma site in the patients analyzed. Despite its rarity in other series, such a frequency was obtained after 10 years of retrospective evaluation in a Brazilian cancer hospital, showing a slightly different frequency in such a population. Among the histological features of ACC, the solid growth pattern is associated with aggressive behavior [6]. Among the clinical features of ACC, a propensity for perineural invasion (PNI) has been reported elsewhere [43], and a systematic review has described PNI as associated with local tumor recurrence [5]. Despite the association of both features with aggressiveness, we did not find a correlation between perineural invasion and solid ACCs in the patients studied. Independent analysis showed 30 miRNAs differentially expressed in solid ACCs compared with tubular and cribriform ACCs, demonstrating a correlation between the histological growth pattern and miRNA expression profiles. Moreover, we reported 49 differentially expressed miRNAs that were able to differentiate patients according to perineural invasion status. These findings provide additional biomarkers of ACC aggressiveness, by considering the miRNAs identified in the solid most aggressive growth pattern and also the PNI status.

The literature review together with the HMDD database evaluation showed that the deregulated miRNAs identified in the solid growth pattern in this study may be involved in salivary gland neoplasms [22], including ACC [19,20]. Zhang et al. [22] reported miRNA expression profiling of salivary gland pleomorphic adenomas compared with matched normal tissue and found that the miRNAs hsa-miR-23b, hsa-miR-154, and hsa-miR-409 were upregulated. Among them, hsa-miR-23b-3p was differentially expressed in both PNI and solid ACCs analyses. This miRNA was also identified to have higher levels in normal salivary gland tissue than in breast tissue [20]. Additionally, the three miRNAs have been described as tumor suppressors [44,45,46,47] and were found to be downregulated in this study. Moreover, downregulation of hsa-miR-23b together with hsa-miR-27b (the miR-23b/27b cluster) was associated with an increase in the risk of disease progression and predicted poor survival in clear cell renal cell carcinoma [48]. Furthermore, the downregulation of hsa-miR-23b was associated with migration and invasion in nasopharyngeal carcinoma and glioblastoma cells [49,50] and was involved in cancer metastasis in prostate, colon, ovarian, hepatocellular, and esophageal squamous cell carcinomas [51,52,53,54]. Similarly, hsa-miR-409 was associated with tumor cell invasion and metastasis in several tumor types [55,56].

Another study by Andreasen et al. [19] reported that the miRNAs hsa-miR-181d and hsa-miR-455 were differentially expressed in the salivary and lacrimal glands compared with normal tissues. miR-181d is a potential tumor suppressor in glioma that regulates important target genes involved in carcinogenesis, such as *KRAS* and *BCL2* [57], and is involved in lymph node metastasis in oral squamous cell carcinoma [58]. The other downregulated miRNA found in our analysis, hsa-miR-455, targets eukaryotic translation initiation factor 4E (*eIF4E*), which affects the translation of selected mRNAs involved in the control of the tumor microenvironment and progression [59].

The five miRNAs downregulated in the solid growth pattern compared with the cribriform and tubular patterns (hsa-miR-23b, hsa-miR-154, hsa-miR-409, hsa-miR-181d, and hsa-miR-455) have been widely associated with tumor suppression, migration, invasion, and metastasis processes in several tumor types, supporting the relationship with the more aggressive behavior of this growth pattern type [44,45,46,47,49,50,51,52,53,54,55,56]. These miRNAs showed a different expression pattern between the major and minor salivary glands. It has been reported that the prognosis of malignant tumors of minor salivary glands could favor tumor recurrence and a worse prognosis [60].

In contrast, six miRNAs previously described in salivary gland neoplasms or ACC (hsa-miR-29c, hsa-miR-140, hsa-miR-195, hsa-miR-24, hsa-miR-143, and hsa-miR-21) were found to be upregulated in patients with perineural invasion. In this analysis, most miRNAs were upregulated in the patients presenting PNI, showing two groups according to the presence/absence of PNI. However, two patients were clustered in different categories. One patient without PNI was previously identified as clustered with the PNI-positive group. The miRNA expression signature could be helpful in this context to identify possible false-negative cases. However, one positive case clustered with the PNI-negative group but was at the interface between the two groups. Among the miRNAs found, miR-21 is currently identified as an ’oncomiR’ that targets several key tumor suppressor genes, such as *PTEN* and *PDCD4*, including the last one in ACC [61]. Extensive studies have described this miRNA’s role during tumor pathogenesis and all other carcinogenesis stages, as reviewed by [62]. MiR-21 is also the most-described miRNA in ACC among all the miRNAs identified in this study [18,22,27,39,40,61,63]. Fujita et al. [64] validated *NFIB* as both a regulator and target of miR-21 in an evolutionarily conserved, double-negative feedback regulatory mechanism. The role of *NFIB* in ACC has been revealed since the truncation of *NFIB* mRNA due to genomic rearrangements can presumably lead to the loss of its function [12]. Further evidence has shown that truncating mutations and homozygous deletions affect *NFIB* function [8], which could directly impact miR-21 expression and several other associated downstream processes. All the other miRNAs found to be expressed in salivary gland neoplasms or ACC (hsa-miR-29c, hsa-miR-140, hsa-miR-195, hsa-miR-24, and hsa-miR-143) (Table 2) [21,22,23,24,25,26] have been previously associated with proliferation, migration, invasion, apoptosis, and/or metastasis mechanisms [65,66,67,68]. This is in line with the pathways found in this study, including pathways related to cancer, such as cell proliferation (Hippo signaling pathway), apoptosis, and tumor progression (TGF-beta signaling pathway). These pathways were also found in the solid tumor growth pattern analysis despite the differences regarding miRNA modulation and targets, reinforcing the possible role of these mechanisms in ACC progression.

In conclusion, we used bioinformatic tools and dedicated databases to identify differentially expressed miRNAs associated with the solid growth pattern and perineural invasion in ACCs. Overall, the current study has some limitations. First, the selection bias associated with the retrospective design in a single center. The small sample size is also an issue, but it is important to consider the rarity of this tumor type. Further investigation is necessary to validate the potential clinical implications of the miRNAs identified that can be associated with the aggressiveness of the disease.

## Figures and Tables

**Figure 1 genes-14-01220-f001:**
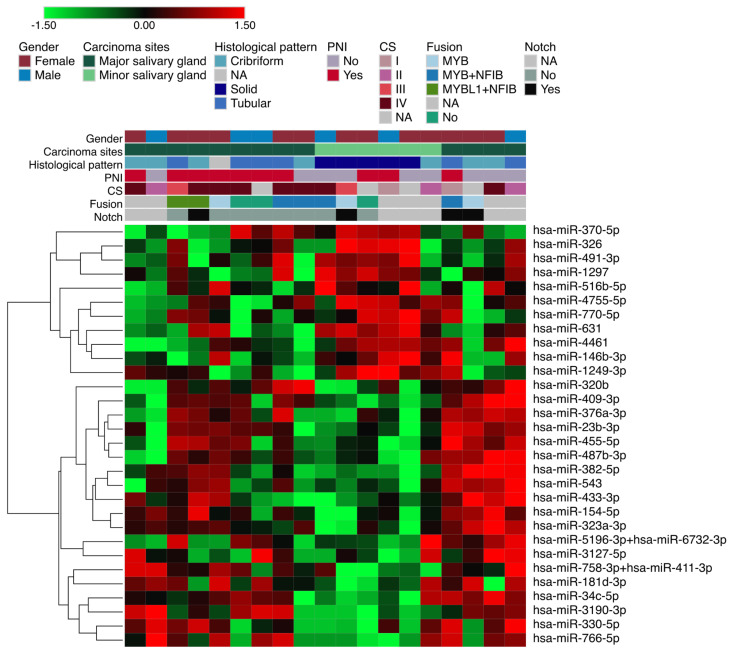
Differentially expressed miRNAs according to growth pattern. The red and green colors indicate up-regulation and down-regulation of miRNAs. The positivity for *NOTCH1* and fusion genes according to the expression levels was determined by the average plus or minus two standard deviations of expression values of all the samples.

**Figure 2 genes-14-01220-f002:**
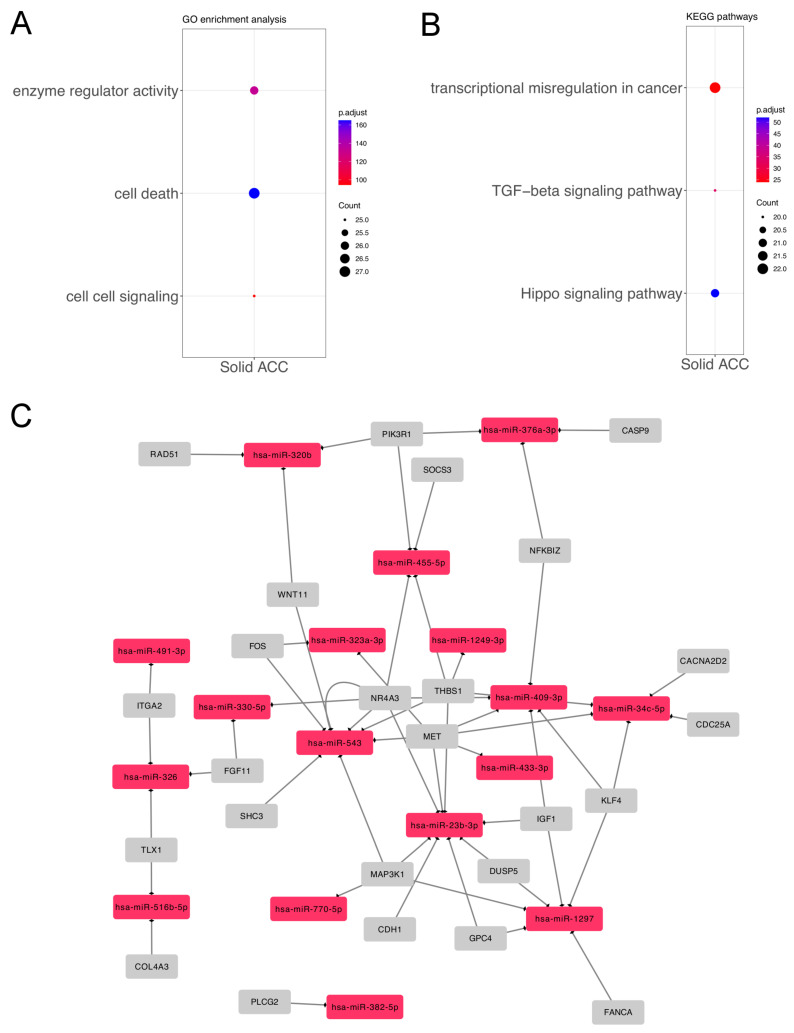
Functional analysis of miRNAs according to growth pattern. (**A**) Top Gene Ontology (GO) categories of solid ACC group; (**B**) Top three KEGG pathways; (**C**) Gene network of miRNA–mRNAs showing targets differentially expressed in PanCancer Pathways panel.

**Figure 3 genes-14-01220-f003:**
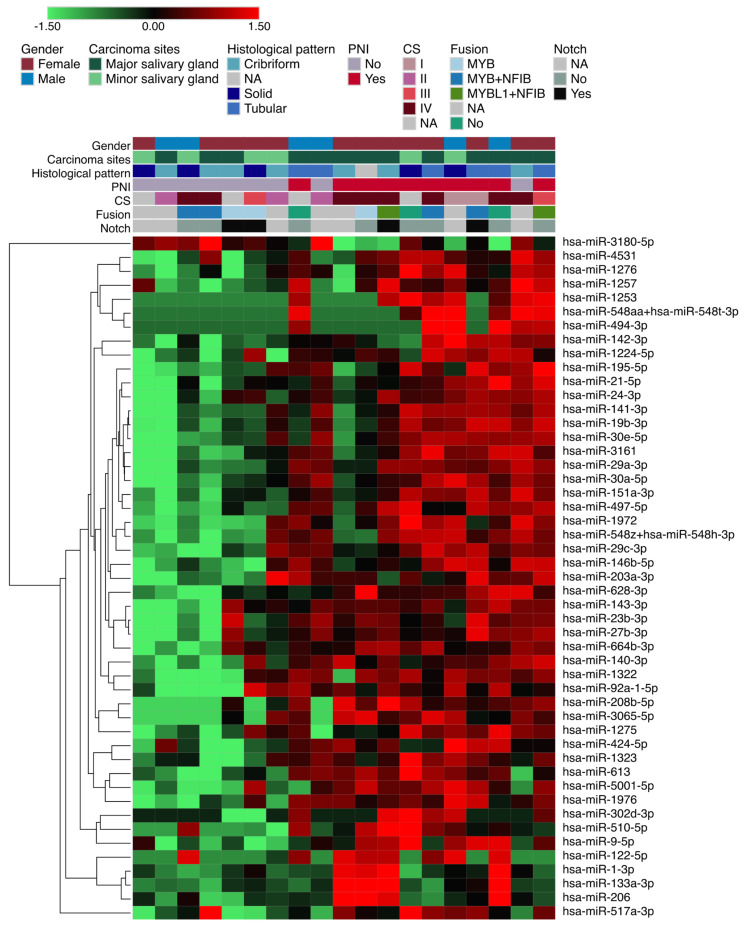
Differentially expressed miRNAs according to perineural invasion (PNI) status. The red and green colors indicate up-regulation and down-regulation of miRNAs. The positivity for *NOTCH1* and fusion genes according to the expression levels was determined by the average plus or minus two standard deviations of expression values of all the samples.

**Figure 4 genes-14-01220-f004:**
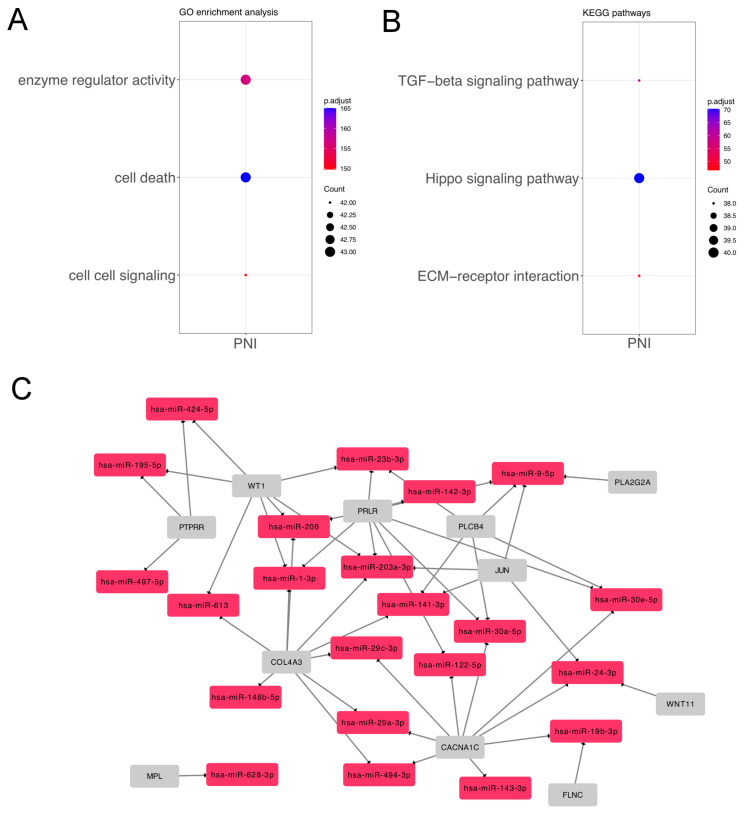
Functional analysis of miRNAs according to PNI. (**A**) Top Gene Ontology (GO) categories of PNI group; (**B**) Top three KEGG pathways; (**C**) Gene network of miRNA–mRNAs showing targets differentially expressed in PanCancer Pathways panel.

**Figure 5 genes-14-01220-f005:**
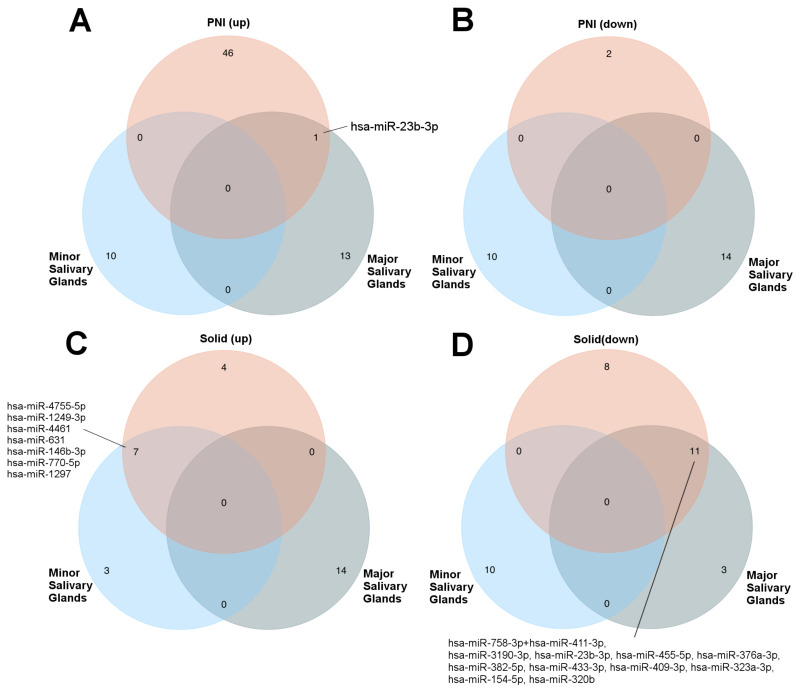
Differentially expressed miRNAs according to growth pattern and perineural invasion (PNI) status related to the site of origin. The specific or shared miRNAs with minor or major salivary glands in each analysis are represented in Venn diagram format. (**A**) miRNAs upregulated in patients presenting with PNI as a comparison group; (**B**) miRNAs downregulated in patients presenting with PNI; (**C**) miRNAs upregulated in solid histological pattern group; (**D**) miRNAs downregulated in the solid histological pattern group.

**Table 1 genes-14-01220-t001:** Clinicopathological features of the patients.

Clinical Data	Category *	Frequency
Sex	Male	6 (31.57%)
	Female	13 (68.4%)
Age	Average (min–max)	54.0 (30–89)
Carcinoma sites	Paranasal sinuses	5 (26.3%)
	Floor of the mouth	3 (15.7%)
	Oropharynx	2 (10.5%)
	Parotid gland	6 (31.5%)
	Submandibular gland	2 (10.5%)
	Base of the tongue	1 (5.2%)
Histological pattern	Cribriform	7 (36.8%)
	Tubular	6 (31.5%)
	Solid	5 (26.3%)
	N/A	1 (5.2%)
Perineural invasion	Yes	11 (57.8%)
	No	8 (42.1%)
Clinical stage	I	1 (5.2%)
	II	5 (26.3%)
	III	2 (10.5%)
	IV	9 (47.3%)
	N/A	2 (10.5%)
Surgery	Yes	19 (100%)
Radiation therapy	Yes	19 (100%)
Chemotherapy	Yes	1 (5.2%)
	No	18 (94.7%)
Follow-up	Alive with disease	4 (21.0%)
	Alive without disease	12 (63.2%)
	Died from the disease	3 (15.8%)

* N/A = not available.

**Table 2 genes-14-01220-t002:** Differentially expressed miRNAs in ACC reported in previous studies.

Analysis	MicroRNA	HMDD Category	References
Solid growth pattern	hsa-miR-181d	Carcinoma, Adenoid Cystic	[19]
Solid growth pattern	hsa-miR-23b	Carcinoma, Adenoid Cystic;	[20]
		Salivary Gland Neoplasms	[22]
Solid growth pattern	hsa-miR-455	Carcinoma, Adenoid Cystic	[19]
Solid growth pattern	hsa-mir-154-5p	Salivary Gland Neoplasms	[22]
Solid growth pattern	hsa-mir-409	Salivary Gland Neoplasms	[22]
Perineural invasion	hsa-mir-29c	Salivary Gland Neoplasms	[22]
Perineural invasion	hsa-mir-140	Salivary Gland Neoplasms	[23]
Perineural invasion	hsa-mir-195	Salivary Gland Neoplasms	[24]
Perineural invasion	hsa-mir-24	Lacrimal Adenoid Cystic Carcinoma	[25]
		Salivary Gland Neoplasms	[21]
Perineural invasion	hsa-mir-143	Carcinoma, Salivary Adenoid Cystic	[26]
Perineural invasion	hsa-mir-21	Carcinoma, Salivary Adenoid Cystic;	[39]
		Carcinoma, Adenoid Cystic	[18]
		Salivaray Gland Neoplasms	[40]

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
