# Peer review of "Identification of MicroRNA Expression Profiles Related to the Aggressiveness of Salivary Gland Adenoid Cystic Carcinomas"

_genes, 2023, doi:10.3390/genes14061220_

Round 1

Reviewer 1 Report

This manuscript by Zanon et al. investigated the miRNA profiles of ACC patients and identified differentially expressed miRNAs associated with different growth patterns and perineural invasion status. The analysis method was adequately described but the overall analysis was too simplistic since all major conclusions were based on the differentially expressed miRNA analysis. More functional implications or validation with other data types could strengthen the conclusions greatly. 

1. Please provide a non-supervised clustered heatmap for all the miRNAs for the dataset. For only <20 samples together, the authors should attempt to see if the overall miRNA profiles could be separated, and try to make some observations based on the dendrogram with clinical features. 

2. In figure 1, the clustering pattern of samples seems to suggest that there are common features in the middle six samples. Why is that? The authors may want to comment on that. And it is curious why the authors selected 2 as the fold change threshold. What if a looser threshold is chosen? Could we see more differentially expressed miRNAs? How would the GO analysis change? And they might give more intersections for Figure 3. 

3. Please consider making some figures for the GO analysis results by showing the statistical significance and gene ratio or enrichment scores. 

4. Is it possible to conduct a regulatory network analysis of some kind for the differentially expressed miRNAs? 

Author Response

We thank the reviewer for the important questions raised. Intending to answer all the points with the functional implications required, we included new figures, a new PanCancer pathways panel experiment, and a supplementary data, as follows:

1) We agree with the reviewer that a non-supervised cluster is highly informative. We performed hierarchical clustering using Pearson's correlation as the measure the similarity (now available as Supplementary Figure 1). The clinical feature description suggested was also included in the Results' section (page 5).

2) The common features showed by the six patients (Figure 1) is related to the solid growth pattern (5 patients) that clusters together with a patient of Cribriform pattern. In the new non-supervised cluster (Supplementary Figure 1), the patients samples can be clustered according to the histological pattern. However, the carcinoma sites also show clear separation. And the patient with Cribiform pattern shares at the same site as the other 5 solid patients (minor salivary gland). Regarding the fold-change (FC) question, we decided to perform a more restrictive analysis since we used a panel of ~700 miRNAs.  A total of 30-50 miRNAs differentially expressed can represent around 5% of the panel, and we focused on them. A less robust FC could provide biased results and does not change the very relevant biological processes that we selected because we selected the top categories. 

3) The Figures requested are now available as Figures 2A/B and 4A/B.

4) We perform a new experiment using a PanCancer Pathways Panel to answer this question. We used this strategy due to a high number of miRNA targets. The networks (now Fig2C and 4C) included only mRNA targets in 13 canonical pathways related to cancer (https://nanostring.com/products/ncounter-assays-panels/oncology/ncounter-pancancer-pathways-panel/).

Reviewer 2 Report

These authors analyzed the microRNA (miRNA) expression patterns in 19 archived samples of salivary gland adenoid cystic carcinoma (ACC). The samples were roughly evenly divided into the three common histological subtypes, cribriform (n=5), tubular (n=6) and solid (n=5).

The authors found some miRs with expression patterns that correlated with carcinoma sites (major or minor salivary glands), histologic pattern and perineural invasion. These are correlations, and the authors provide no evidence that the miRs play any role in these phenotypes. None of the miRs correlated with expression of MYB or with mutations in NOTCH (the latter were apparently not investigated). Furthermore, the miRs do not appear to be very useful biomarkers since they do not provide any more information beyond what a skilled pathologist provides by looking at the tissue sample.

Although there is nothing major wrong with this paper, the cohort size is very small and the findings do not really move the field forward at all. The authors claim that their findings “provide additional biomarkers of ACC aggressiveness”, but they would need to validate their findings with an independent cohort of samples to make that claim. There are much larger studies of both miRs (refs 15 and 16) and mRNAs (not cited: PMIDs 36900183 and 29484115) in ACC tumors – the authors did not really make an effort to explain their results in the context of the previous studies. Why did the authors not mine the previously published data to validate their findings?

The authors have not provided enough detail about their analysis of MYB and NOTCH in these samples – did they only analyze expression of the transcripts or did they also check for fusions and/or mutations? There is no description of the latter in the Materials and Methods, but Figures 1 and 2 list several categories of MYB fusions and NOTCH (NA, No, Yes: are these mutations?), without any explanation. (The figure legends have no information.).

Author Response

We thank the reviewer for the very deep evaluation of our work. However, we did not study NOTCH mutations but the gene expression. To avoid this misunderstanding, the description of the custom panel was expanded (page 4). describing the changes in expression levels of MYB, NOTCH, and other transcripts. 

  • Regarding the cohort question, it is small because we selected ACCs from the salivary gland, which is rare. We obtained only 19 patients with all the requirements and quality for the analysis between 1998 and 2012 (the time of the study approval). This information is on Methods's page 2.
  • We thank the reviewer for the suggestion of literature articles. We focused mainly on the publications provided by the HMDD database, and we apologize for this. We included these other references in this new version. 
  • Regarding the validation, we do not have another cohort to test, considering the rarity of the tumor. However, we included the information from the HMDD database reporting the identification of our differentially expressed miRNAs in other similar studies in ACC, reinforcing our findings (Table 2). 

Round 2

Reviewer 2 Report

The authors made some very minor changes to the manuscript to address my previous comments, but did not address these issues that were pointed out previously:

1. The authors found some miRs with expression patterns that correlated with carcinoma sites (major or minor salivary glands), histologic pattern and perineural invasion. These are correlations, and the authors provide no evidence that the miRs play any role in these phenotypes. The authors should explain why their results are important (if they are). Are the miRs just surrogates for cell type analysis? Or are they linked to clinical outcomes?

2. None of the miRs correlated with expression of MYB or with mutations in NOTCH (the latter were apparently not investigated). Metastases of ACC has been associated with mutations (not expression) of Notch1, so the discussion about Notch1 is still confusing. The authors should add the word "expression" to the legend at the top of Figures 1 and 3 so it is clear that they are only reporting the expression of the MYB/MYBL1/NFIB and Notch1 genes and not mutations. (Also: be consistent and use Notch or Notch1 but not both.)

3. Furthermore, the miRs do not appear to be very useful biomarkers since they do not provide any more information beyond what a skilled pathologist provides by looking at the tissue sample (e.g. PNI). If there are advantages to using the microRNAs as markers instead of having a Pathologist judge the PNI, the authors should clearly explain that.

4. The authors claim that their findings “provide additional biomarkers of ACC aggressiveness”, but they would need to validate their findings with an independent cohort of samples to make that claim. The text should be changed to indicate that these findings need to be validated before they can be used as biomarkers.

Author Response

We provide a point-by-point answers, as follows:

1. Studies on gene/microRNA expression describe patterns and correlations and does not mean they are not important. Regarding the context of the correlations evaluated, there are evidences in literature that solid histologic pattern is the most aggressive. Furthermore, perineural invasion (PNI) can be found in a variety of malignant tumors, also considered as a sign of tumor metastasis and invasion and the poor prognosis of patients.  The goal of the present study is the identification of miRNA expression profiles related to the aggressiveness.  It is a screening study and this information is provided in the last sentences of abstract and introduction.

2. A new reference was added to the manuscript. Ferraroto et al. 2017 showed that NOTCH1-mutant tumors presented higher levels of Notch1 pathway activation (gene expression and protein levels), and also define a distinct aggressive ACC subgroup. We added the "expression" word to the legends, as suggested. We used NOTCH1 to refer to the expression of the gene and Notch to the signaling pathway, following HUGO gene nomenclature and KEGG pathway standard. 

3. The goal of the present study is the identification of miRNA expression profiles related to the aggressiveness. A new sentence was added in the discussion to clarify that.

4. This information was provided and it is highlighted in the discussion section.